# Functional redundancy compensates for decline of dominant ant species

Peter Yeeles [1,2] ✉, Lori Lach[2], Richard J. Hobbs[1] & Raphael K. Didham [1,3] ✉

Evidence is accumulating of declines in widespread, abundant insect species. The consequences of these losses for ecosystem functioning are predicted to be severe but remain poorly tested in real-world ecosystems. Here we tested the relative importance of functional redundancy versus complementarity in conferring stability of multifunctional performance in the face of dominant insect species decline. We conducted an experimental manipulation of functional trait-space occupancy within naturally occurring ant communities in Australia. Experimental suppression of dominant ant species in multiple trait groupings caused a counterintuitive increase in multifunctional performance, which was associated with an increase in species richness. The resident ant community had high functional redundancy, contributing to rapid compensatory dynamics following suppression. However, colonization by new species with increased trait complementarity drove higher multifunctional performance. This increased multifunctionality probably occurred via reduced interspecific competition but at the cost of increased sensitivity of ecosystem multifunctionality to further species loss. Our findings show that functional redundancy can buffer multifunctional performance of a community against decline of dominant insect species but suggest that future stability of ecosystem multifunctionality depends more on functional complementarity and altered competitive interactions.

Dramatic declines in the abundance and biomass of terrestrial insects have been reported across many regions of the world[1–3]. Alarmingly, some of the greatest declines have been observed in formerly abundant, widespread insect species[4–8], with important ramifications for ecosystem functioning[9]. Although biological diversity in one form or another underpins the performance of ecosystem functions[10–12], evidence suggests it is the abundance of common species in particular that maintains ecosystem function and drives delivery of ecosystem services[13] despite non-random loss of species[14]. Given the rapid decline in common insects, it is crucial to understand the resilience of communities to loss of the most abundant species so as to mitigate potential declines in ecosystem services.

In biodiversity–ecosystem function theory, functional redundancy is thought to be an important mechanism supporting ecosystem function in the face of fluctuating diversity[15,16]. Evidence for functional redundancy hinges on the observation that functional diversity (for example, morphometric and life history traits) typically increases asymptotically with increasing species richness[17]. Therefore, species-rich communities are more likely than depauperate communities to contain multiple functionally similar species that have the potential to buffer decline of common species. However, empirical tests of functional redundancy within biodiversity–ecosystem function relationships have focused primarily on manipulation of plant diversity[18] rather than on insects, and they have often only addressed the performance of single functions, such as primary productivity[19]. Functional redundancy theory is commensurately more challenging to test across multiple functions and across multiple producer and consumer trophic levels[20–23], because consumer species are likely to have many

[1]School of Biological Sciences, University of Western Australia, Crawley, Western Australia, Australia. [2]College of Science and Engineering, James Cook University, Cairns, Queensland, Australia. [3]CSIRO Health and Biosecurity, Centre for Environment and Life Sciences, Floreat, Western Australia, Australia. ✉e-mail: peter.yeeles@jcu.edu.au; Raphael.Didham@csiro.au

more trait–functional dimensions of importance in ecosystem service provision. Moreover, multiple species that are 'redundant' in one trait–functional dimension may be highly complementary in another. Some authors have suggested that a species with high trait redundancy in a community (that is, one with many functionally equivalent analogues in the system) may only realistically be redundant for single functions[24,25]. For example, in ants, a generalist species that can consume seeds and scavenge may be functionally redundant in the presence of a specialist granivore with some similar traits but not multifunctionally redundant if that specialist is unable to also scavenge to a similar performance rate. This 'low multifunctional redundancy' and high complementarity limit interspecific competition and may be important drivers of multifunctionality[26]. Testing whether higher complementarity leads to higher multifunctionality would provide important insight into whether consumer communities are less resilient to dominant species loss when considered from a multiple-function perspective[25].

Here we present a test of the importance of functional redundancy for multifunctional performance in higher-level consumers, using species removal experiments[27] to manipulate ant community structure through suppression of the most abundant and widespread species within multiple functional trait groupings. Ants are an ideal candidate taxon for this purpose owing to their breadth of ecological roles, social nature and stable foraging ranges. Ant removal experiments have been used many times and in differing configurations for testing the effects of competition[28,29], community assembly theory[30], invasion biology[31] and functional importance[32–34]. First, we characterized functional trait space for the naturally occurring ant community at the Ridgefield TreeDivNet[35] experimental tree diversity site in southwestern Australia[36] on the basis of ten traits for 26,812 individuals of 34 species captured in 400 pitfall traps. We used a trait-based approach to define functional groupings, because species are more likely to perform similar functions if they have similar traits. Analysis of functional dispersion in trait space identified five nominal trait groupings, and, within each grouping, we identified the numerically dominant 'target' species for suppression (that is, the species with the highest site-wide incidence), three of which proved to be logistically feasible to manipulate following a methodological trial (Extended Data Fig. 1). We experimentally suppressed dominant species in a replicated split-plot design of paired control and suppression plots (Extended Data Fig. 2). The ant suppression treatment was successful in removing colonies of the three species that were numerically dominant in each of their respective functional trait groupings, *Iridomyrmex purpureus*, *Pheidole ampla perthensis* and *Tetramorium impressum*. We were also successful in suppressing incipient new colonies across the full year of the experiment. We observed reductions in abundance of 94% for *I. purpureus*, 96% for *P. ampla perthensis* and 99% for *T. impressum*, relative to baseline capture rates in suppression plots, although long-range foragers and workers from declining or incipient colonies still occurred in some pitfall traps in suppression plots. We tested whether suppression of the dominant member from each identified functional trait grouping (Extended Data Fig. 1) would lead to (1) a reduction in the performance rate of each of four single functions, measured using field observations; (2) a decline in overall multifunctional performance rates via cascading direct and indirect effects on species richness and evenness; and (3) changes to interspecific competitive hierarchies resulting in previously rare ants increasing in abundance and functional contribution, relative to the remaining ant community (that is, abundance turnover).

## Results and discussion
### Community-wide responses to dominant ant suppression
Overall, after accounting for non-independence of replicates in the split-plot experimental design using a mixed-effects modelling framework, we found that suppression of the three species had a significant positive, not negative, effect on non-target ants (that is, species other than the three manipulated target species). The abundance

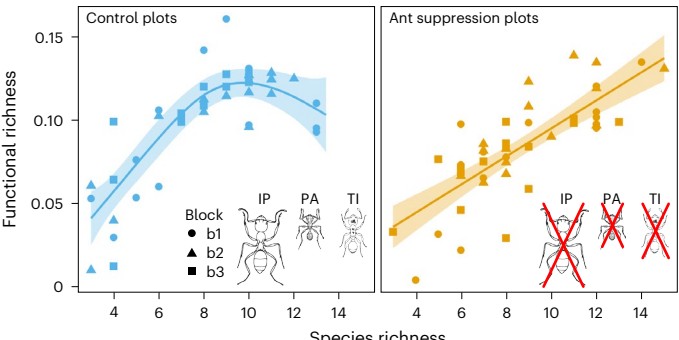

**Fig. 1 | Suppression of the dominant members of three functional groupings resulted in linearization of the SR–FR relationship.** Relationship between species richness and functional richness in control and ant suppression zones, fitted using a GAMM incorporating the random effects of plot nested within spatial block to account for repeated measurements per plot within the split-plot design. Solid lines represent the estimated mean value from the best-fit GAMMs, selected by comparison of AIC values among competing models (Supplementary Table 4). Shaded ribbons represent 95% confidence intervals. Data include all non-target and target species in the suppression experiment. The three ant species that were suppressed and their relative sizes are shown: IP, *I. purpureus*; PA, *P. ampla perthensis*; TI, *T. impressum*.

(generalized linear mixed model (GLMM), $Z = 2.60$, $P < 0.05$), richness (GLMM, $Z = 2.79$, $P < 0.05$) and effective number of species (ENS) (linear mixed model (LMM), $Z = 2.28$, $P < 0.05$) of ants in pitfall traps all increased in ant suppression plots after 1 year. However, we found no significant effect of suppression on ant compositional uniqueness (LMM, $Z = 0.87$, $P = 0.39$), functional richness (LMM, $Z = 1.30$, $P = 0.18$) or functional dispersion (LMM, $Z = 1.30$, $P = 0.16$) of non-target ants.

One signature of functional redundancy is asymptotic saturation of functional richness with increasing species richness[37]. Comparisons of model fit using generalized additive mixed models (GAMMs) showed that species richness versus functional richness (SR–FR) relationships differed significantly between treatments (ΔAIC (Akaike information criterion) values are shown in Supplementary Table 1). The SR–FR relationship in control plots was strongly nonlinear, approaching an asymptote (GAMM, estimated degrees of freedom (e.d.f.) = 2.905, number of basis functions $k' = 9$, $F = 31.36$, $P < 0.001$; Fig. 1), whereas the relationship in ant suppression plots was approximately linear (GAMM, e.d.f. = 1.002, $k' = 9$, $F = 68.25$, $P < 0.001$; Fig. 1). Although communities in both control and suppression plots attained approximately the same maximum values of functional richness, control communities did so at lower species richness (Fig. 1). The significant nonlinearity in the SR–FR relationship in control plots indicated the presence of functionally redundant species in plots with greater species richness. This was notable, as although saturating relationships are commonly predicted—for example, in ref. 38—in reality, linear relationships are more often reported in studies of consumer level taxa, for example, in ants[39], beetles[40] and birds[41]. The significant difference in SR–FR relationships between suppression and control plots showed that we had successfully reduced the degree of functional redundancy in these communities. The saturation of the SR–FR relationship in naturally assembled communities is thought to be indicative of a random assembly process, with a steepening of the relationship indicating overdispersion (for example, that resulting from competitive effects) or linearization suggesting underdispersion (from selection pressures such as environmental filtering)[37,42]. This could mean that local competitive effects were important in defining community structure throughout the course of our experiment, whereas the linear rather than asymptotic SR–FR relationship following dominant species suppression (Fig. 1) might indicate lessening of interspecific competitive effects.

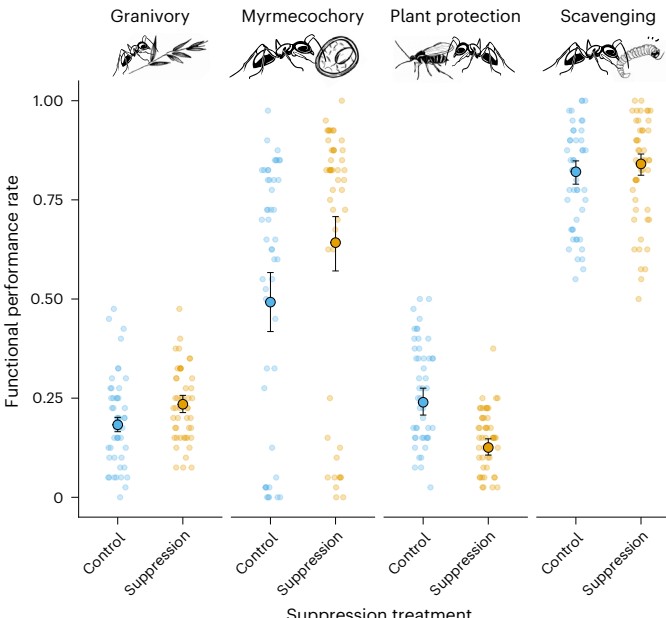

**Fig. 2 | Suppression of the dominant members of three functional groupings resulted in a significant increase in the rate of myrmecochory, a significant decrease in plant protection, and no change to rates of granivory or scavenging.** Small points represent the raw data ($n = 48$ for each treatment by function combination, based on four sampling events in two plot quarters nested within each of two plots within each of the three spatial blocks in the experiment). The larger points show the estimated mean values from generalized multilevel confirmatory path analyses built using the 'piecewiseSEM' R package[67], accounting for non-independence in the data using random effects for spatial block and plot nested within block. See Fig. 3 for fitted path models and statistical significance of suppression effects on each function. For illustrative purposes, error bars show 84% confidence intervals; non-overlapping confidence intervals represent situations in which mean values are significantly different at the $P < 0.05$ level (see Supplementary Note 1 for further details).

## Contrasting effects on multiple ecosystem functions

In structural equation models partitioning the direct versus indirect effects of ant suppression on shifts in biodiversity–ecosystem functioning relationships, we found a significant positive effect of ant suppression on the overall rate of granivory and a significant negative effect on plant protection (Fig. 2), whereas there was no direct effect on myrmecochory or scavenging rates. We found that responses were predominantly mediated by increases in non-target ant species richness following suppression of dominant species. For example, our structural equation models showed that whereas there was no independent direct effect of ant suppression on myrmecochory when species richness was held constant at its mean value (standardized partial effect size: $\beta = -0.014$, $P = 0.86$), there was a significant positive indirect effect of richness on function ($\beta = 0.585$, $P < 0.001$). Granivory performance rate was negatively associated with increasing species richness ($\beta = -0.536$, $P < 0.001$, Fig. 3a), but this effect was moderated by an interaction between treatment and richness ($\beta = 0.313$, $P = 0.023$), with the negative biodiversity–ecosystem function response becoming weaker in the ant suppression zones. Essentially, at higher species richness levels, the positive effect of dominant ant suppression on granivory became progressively stronger. Rates of plant protection were positively driven by the increase in species richness ($\beta = 0.492$, $P < 0.001$), but this was masked by a large independent negative effect of dominant ant suppression on function when species richness was held constant at its mean value ($\beta = -0.576$, $P < 0.001$, Fig. 3a). There were no direct or indirect effects of dominant ant suppression on scavenging rates.

The observed diversity–function relationships mediating the effects of ant suppression on each of the single functions still held true when we considered variation in the relative abundances of species, rather than just species presence or absence. Path models incorporating the ENS showed qualitatively similar relationships to those obtained with the richness models, albeit with smaller effect sizes (Extended Data Fig. 3), suggesting that variation in relative abundance does not alter conclusions about multifunctionality.

The contrasting responses of individual functions (Figs. 2 and 3) meant that plot-level covariance across functions was not uniformly positive (Extended Data Fig. 4). For instance, in the control treatment, there was positive covariance in measures of myrmecochory, plant protection and scavenging but significant negative covariance between granivory and plant protection and between granivory and myrmecochory (Extended Data Fig. 4). These associations were similar in suppression treatment plots, except that the shift in the biodiversity–ecosystem function relationship for granivory seen in the path models (Fig. 3a) resulted in a much-weakened negative covariance between granivory and myrmecochory and a non-significant relationship between granivory and plant protection.

Negative covariance among rates of single-function performance is common in multifunctionality datasets[23,26,43] and could be seen clearly when comparing rates of our four measured functions. Potentially, this could be seen as a trade-off in which high performance of one function precludes a similarly high rate in another. However, we can see no inherent biological reason that high assemblage-level performance of one function, such as granivory, should mean that other functions, such as plant protection, cannot be performed at a similar level. It is more likely that negative covariance reflects changes in resource availability and associated species preferences under differing levels of indirect or direct competition. For example, it could be expected that ants would prefer high quality and low effort resources, such as invertebrate carrion, but their access to these resources depends on local competitive effects. Despite evidence for negative covariance among functions, effective multifunctionality still increased in the ant suppression treatment but was predominantly mediated by an increase in species richness, with no residual direct effect of the suppression treatment when species richness was held constant at its mean value. This effect still held when we accounted for variation in species abundances using the ENS, although both the effect of treatment on the ENS and the consequential effect on effective multifunctionality were smaller (Extended Data Fig. 3).

Although there was no evidence that the slope of the biodiversity–multifunctionality relationship varied between treatments in the effective multifunctionality ($q = 1$) analysis (that is, no significant interaction effect, as shown in Fig. 3b), and no residual direct effect of treatment, the effective multifunctionality approach masked substantial variation in diversity effects at differing functional thresholds (Fig. 4). The threshold analysis evaluated whether diversity became increasingly important in the simultaneous performance of multiple functions above a specified threshold level. In multiple-thresholds analyses, diversity always had a positive effect on multifunctionality (that is, 'diversity effect' values greater than zero in Fig. 4), but the biodiversity–multifunctionality effect was weaker in control plots with higher negative covariance among functions, and stronger in the ant suppression plots. For instance, the statistically significant biodiversity–multifunctionality effect (that is, where the 95% confidence interval did not overlap with zero in Fig. 4) occurred over a narrower range of functional thresholds in control plots (78% to 81%) compared with ant suppression plots (63% to 84%) and with a lower maximum slope (realized maximum diversity effect ($R_{mde}$) = 0.09) than in suppression plots ($R_{mde}$ = 0.12; that is, the predicted increase in multifunctionality per species added to the system). These $R_{mde}$ values equated to a predicted total of 11.1 species required to perform one additional function above that threshold in control zones, but only 8.3 species were required for

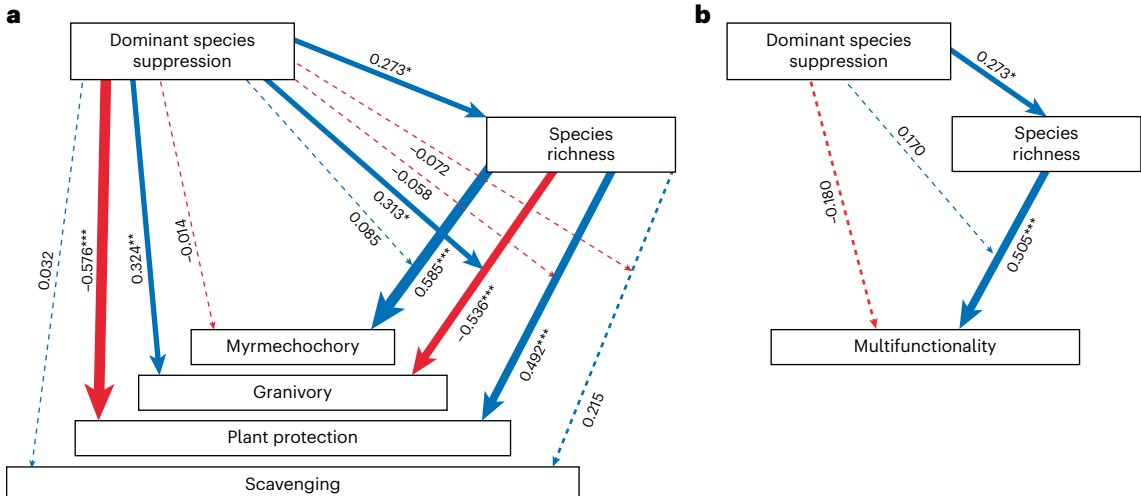

**Fig. 3 | Functional rate responses following ant suppression were predominantly mediated by increases in non-target ant species richness. a,b**, Path models built using generalized multilevel confirmatory path analyses in the structural equation model package 'piecewiseSEM' in R[67] showing the relationships between dominant species suppression and ecosystem function, as mediated by species richness (**a**); and dominant species suppression and multifunctionality, as mediated by species richness (**b**). Multifunctionality was calculated using the effective multifunctionality ($q = 1$) approach of Byrnes et al.[57]. Species richness was calculated after removal of the target species from each dataset. Arrow widths are scaled by the standardized partial regression coefficient (that is, effect size, as indicated by the values next to paths) and coloured to represent the direction of effect (red arrows show negative effects, blue arrows show positive effects) and are solid rather than dashed for statistically significant effects (*$P < 0.05$, **$P < 0.01$; ***$P < 0.001$). Note that **a** shows a composite of multiple structural equation models, each of which was estimated separately for each individual function (see Supplementary Table 5 for statistical support for standardized and unstandardized model coefficients, 95% confidence intervals, and degrees of freedom).

a comparable effect in treatment zones. Multiple-thresholds analysis using ENS as the diversity measure showed no significant diversity or treatment effects. Testing for an effect of our suppression treatment at the mean threshold of $R_{mde}$ in each of our two multiple-thresholds analyses revealed a significant effect of treatment on the number of functions performed at that threshold (GLMM, $Z = 2.17$, $P = 0.02$).

The species most strongly associated with high functional rates in the control plots varied among individual functions (Extended Data Fig. 5 and Supplementary Table 2), with a total of 12 significant associations (standardized effect size > 1.96). For example, *Iridomyrmex suchieri* was significantly associated with the rate of plant protection (linear model (LM), $Z = 3.21$, $P < 0.05$), but not scavenging (LM, $Z = 0.58$, $P = 0.56$). Many of the 12 species showed substantial shifts in their degree of association with functional rates following suppression of dominant ants (Extended Data Fig. 5). Moreover, these shifts in species' relative associations with function were not idiosyncratic but instead were strongly negatively structured by ant suppression in the presence of dominant species. Species with low standardized effect sizes in control plots (that is, those strongly negatively associated with functional rates in the presence of dominant competitors) tended to have higher functional associations in suppression plots, whereas those species of greater functional importance in control plots were less strongly associated with functional performance in suppression plots (Extended Data Fig. 5). Spearman's rank correlations indicated significant negative associations between species functional importance and the magnitude of response to ant suppression for scavenging ($\rho = -0.58$, $P < 0.05$), granivory ($\rho = -0.69$, $P < 0.05$) and plant protection ($\rho = -0.42$, $P < 0.05$) but not for myrmecochory ($\rho = -0.11$, $P = 0.56$); see Extended Data Fig. 5.

## Insights into the potential consequences of dominant species loss

The pattern of reduced functional redundancy combined with increased functional performance in our ant suppression plots may indicate that shifts in interspecific competitive hierarchies drive changes in functional performance rates. Selection effects, such as the probability of a community containing highly functioning dominant species, are thought to be a process by which richer communities are better able to perform ecological functions[44,45]. However, this may not be the case for all communities, especially when considering multiple rather than single functions. We found increased multifunctional rates in areas where dominant species had been removed, suggesting that selection effects (for dominant species) can have negative implications for overall functional performance. Here, dominant species suppressed the functioning of the resident community, which had higher 'functional potential' when released from the effects of competition. Essentially, the benefit of a community containing highly functioning dominants is conditional and can be outweighed by a loss in functional performance of the remaining community due to competitive suppression. These results potentially support a concept of 'low multifunctional redundancy', a term coined in 2013 by Mori and others[46]. According to this paradigm, which represents an alternative hypothesis to a strictly trait-based concept of redundancy, the maintenance of multifunctionality requires low within-community duplication of functional types and hence higher multifunctional complementarity of species; thus, the stability of ecosystems is derived not from functionally redundant species but from low redundancy with high response variability and spatiotemporal complementarity of species[46–48]. This last point may be further supported by our finding of increased species richness in response to our treatment, suggesting that gaps in niche space left by ant suppression had been taken up by new arrivals from the regional species pool.

The spatial scale of the communities in question is likely to be important in determining whether the detected resilience to functional decline (with non-random species loss) is due to internal redundancy effects, such as the functional upregulation of competitively suppressed species, or the infill of niche space by new arrivals from a regional species pool[47]. Our experimental site, embedded within a matrix of adjacent agricultural land and remnant woodland, probably showed resilience to our suppression treatment through a combination

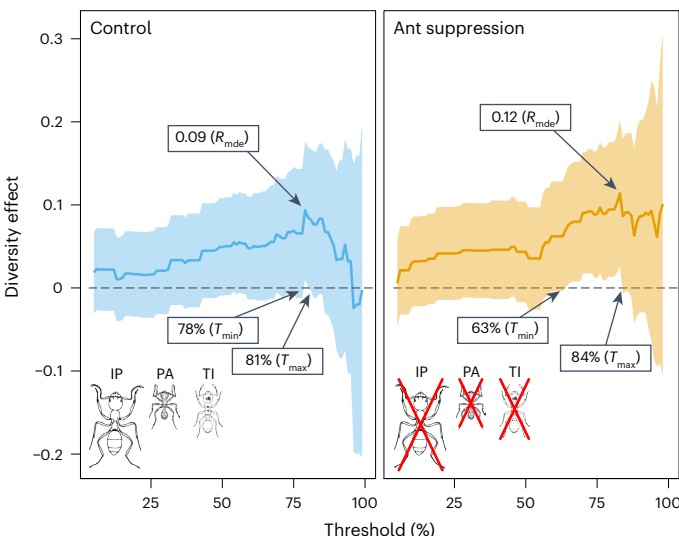

**Fig. 4 | Diversity had a positive effect on multifunctionality, but the biodiversity–multifunctionality effect was weaker in control plots with higher negative covariance among functions and stronger in the ant suppression plots.** Comparison of biodiversity effects against functional performance thresholds. Solid lines representing the diversity effect indicate the estimated mean coefficients (±95% confidence limits) from regressions of species richness versus the number of functions performed above a threshold level (from 5% to 95%). Diversity effect values greater than 0 represent positive biodiversity–multifunctionality relationships. Those that are significant (with 95% confidence intervals that do not overlap zero) represent the predicted increase in multifunctionality per species added to the system. Shaded areas show 95% confidence intervals. $T_{min}$ and $T_{max}$ indicate the breadth of threshold ranges with a significant diversity effect, whereas $R_{mde}$ indicates the diversity effect at its strongest point in the relationship. The three ant species that were suppressed and their relative body sizes are shown.

of these factors. First, our experimental manipulation resulted in increases in both species richness and abundance of other ant species, suggesting that there was some degree of immigration from the surrounding matrix. Second, our species-level analyses showed that original residents that responded most strongly to treatment were those that had the strongest negative associations with functional performance in control areas. This indicates that these species were potentially released from the effects of competitive suppression, either directly through a reduction in foraging competition or indirectly through changes in resource availability, and that they may have upregulated their functional performance to make use of newly available resources. In addition, in treatment zones, multiple-thresholds analysis suggested that where dominant species were removed, each remaining species may have made greater contribution to multifunctionality.

Of course, 'redundancy' as described above should not be taken with the negative connotations of the colloquial term ('superfluous to requirements'), and we make no attempt to imply the absolute redundancy of species per se. In this context, we consider redundancy as a positive buffering effect that increases the reliability of system processes and potentially safeguards against negative consequences of non-random species loss[49]. Species are likely to have many direct and indirect effects on function and interaction networks, and even the most complex field experiments are likely to provide only crude approximations of the true relationship between species diversity and overall ecosystem health[50]. See Supplementary Note 2 for further discussion of species-level responses to treatment.

## Conclusion

Understanding the consequences of changes to biodiversity–multifunctionality relationships in response to the decline of widespread,

abundant insect species will enable the development of effective conservation practices to ensure that both species diversity and ecosystem-wide functional performance are preserved. We have shown that numerous factors, both direct and indirect, play a part in defining the resilience of ecological functioning to perturbation. The presence of functionally redundant species does not always result in more reliable performance of ecosystem functions and processes, especially when one considers complex multifunction scenarios as would be commonly found in nature. Importantly, we found that the loss of ubiquitous dominant species could lead to unexpected increases in multifunctional performance rates where similar competitors are able to upregulate their functional performance. Our findings demonstrate a need for insect decline research to move on from simply documenting changes in abundance or biomass to more targeted investigations of the consequences of biodiversity declines (or increases) for the functioning of ecosystems and the natural ecosystem services on which human welfare depends.

## Methods
### Experimental design

The ant suppression experiment was conducted at the Ridgefield tree diversity site located in southwestern Australia[36] (Extended Data Fig. 2). Ridgefield was designed to investigate how woody plant species diversity influences the provision of multiple ecosystem services, especially in the context of woodland ecosystem restoration, and is part of the TreeDivNet global network of tree diversity experiments[35]. After 4 years of monitoring ant community structure (2010–2013) in the 100 tree diversity plots (21 × 23 m), Yeeles et al.[51] found that there was no significant effect of different woody plant treatments on ant richness or composition, but that community structure instead varied spatially (and idiosyncratically) among plots. Therefore, for the purposes of our ant suppression experiment in 2014–2015, we arbitrarily divided the Ridgefield site into three large spatial blocks that each covered the full range of plot-level variation in species richness. Within each block, we allocated one ant suppression treatment zone and one control zone in a split-plot design, with the requirement that each suppression zone be as far as possible from control zones to avoid unintended 'spillover' of treatment impacts. Owing to the large spatial scale of the suppression areas (relative to ant home range size), it was not possible to remove dominant ant colonies from the entire suppression zone; therefore, our treatments were centred around specific 'focal plots' (of 21 × 23 m, established as part of the Ridgefield tree diversity experiment) and implemented at the scale of the maximum recorded foraging distance for our target species. In each of the six control or suppression zones, there were two focal plots in which all measurements were taken (and around which the ant removals were conducted in the suppression treatment). These plots (12 in total) were planted with *Eucalyptus* trees only and used the same standardized Ridgefield tree treatments in each zone (one plot with 110 *Eucalyptus loxophleba* ssp. *loxophleba* trees and the other with 96 *E. loxophleba* ssp. *loxophleba* and 14 *Eucalyptus astringens* trees[36], selected to maximize the spatial distance between plots within and between spatial blocks).

### Selecting the functionally dominant ant species for suppression

Ant suppressions were carefully designed to test the effects of functional redundancy on multiple ecosystem services. We defined the functional trait grouping structure of the resident ant assemblages and then removed the dominant member of each group, with the assumption that the remaining species in each group would be of a reasonably similar trait-functional type to those species that were suppressed. We therefore expected the remaining species to compensate for the loss of these dominant members and maintain a similar level of function to that observed before treatment imposition. With careful selection of traits that are associated with measurable functions, this

would result in a manipulation of the SR–FR relationship, reducing the degree of redundancy and allowing direct testing of the claim that functional redundancy can add ecological resilience by buffering against non-random species loss.

To select species for suppression, we first collated ant community data obtained from 400 pitfall traps placed across the entire site in November 2012. We captured a total of 26,812 ants from 13 genera and 34 species[51] and calculated incidence rates for each species in the suppression treatment blocks. For each species, we recorded ten phenotypic traits (Supplementary Table 3) that have been demonstrated to have some influence on ant mode of functioning and persistence in the environment[52]. With these traits, we used the functional diversity 'FD' package[53] in R to calculate functional dispersion and ran a principal coordinates analysis to reduce our ten trait dimensions to two that described 36% and 19% of total variance, respectively (Extended Data Fig. 1 and Supplementary Table 4). Species were weighted by their site-wide incidence to ensure that the distribution of traits reflected species occurrence in each block. In the functional diversity analysis, five nominal trait groupings were identified as best representing the distribution of traits at the site; on the basis of these, we identified one dominant 'target' species (that is, the species with the highest site-wide incidence) for the three functional trait groupings that our methodological trial showed were feasible to manipulate (see Supplementary Note 3 for details of the methodological trial). The species selected for suppression were *I. purpureus*, *P. ampla perthensis* and *T. impressum*, all of which were targeted for removal from suppression treatment zones in all three blocks.

## Colony removal

In May 2014, 3 months before suppression was due to begin, we established the mean foraging distances of the three target species by placing shortbread cookie baits in and around the experimental plots and following foraging ants as they returned to their respective colonies after collecting the baits. We found that both *P. ampla perthensis* and *T. impressum* had short foraging distances of 1.31 m (s.e. = 0.14, $n = 32$) and 0.90 m (s.e. = 0.11, $n = 17$), respectively, whereas *I. purpureus* foraged farther at an average of 6.58 m (s.e. = 1.03, $n = 45$), with a maximum recorded distance of 42 m.

Ant colonies were removed using a combination of insecticide inundation, mechanical destruction and colony-specific barriers into which we provided vials of an ingestible poison. A first round of colony removals took place throughout the (winter) month of August 2014 (August colonies destroyed: *I. purpureus*, 30; *T. impressum*, 17; *P. ampla perthensis*, 27). An additional colony mapping and removal exercise was carried out in late September (spring) as soil temperatures increased and more colonies became surface-active (September colonies destroyed: *I. purpureus*, 2; *T. impressum*, 6; *P. ampla perthensis*, 9).

Nest inundation was carried out by funnelling deltamethrin-based insecticide (10 g l$^{-1}$ deltamethrin, Delforce, Sherwin Chemicals) mixed with water (per the product instructions) directly into nest entrances[54]. For *T. impressum* and *P. ampla perthensis* colonies, approximately 1 l of insecticide was used. For *I. purpureus* colonies, the quantity of insecticide used was proportionate to the diameter of the colony, with approximately 5 l used on colonies of less than 50 cm diameter and up to 15 l used on colonies of a greater size. The volume of diluted insecticide was recorded, and an equivalent volume of water was introduced haphazardly around the centre of each control plot. After 24 h, each nest was mechanically destroyed to a depth of approximately 40 cm using a 15-mm-diameter steel rod, with an equivalent disturbance carried out haphazardly in control plots. The site was checked every 6 weeks for the duration of the experiment to ensure successful suppression of the target species. We treated a further 26 newly established colonies of *I. purpureus* throughout the experiment using a barrier and bait system that allowed no non-target effects during the measurement period. This entailed a 1-week placement of poison bait adjacent to the entrance of the colony to be treated, with both the bait and entrance covered with an upturned bucket that was dug into the ground to a depth of 5 cm, ensuring that only ants in the target colony could access the bait. Over the year of the experiment, a total of seven incipient colonies of *P. ampla perthensis* and *T. impressum* were treated using localized nest inundation and mechanical destruction.

## Ant community responses to suppression of dominant species

To assess ant community responses to target species suppression, we sampled across the site every 3 months for 1 year following treatment imposition: November 2014, February 2015, May 2015 and November 2015. We used pitfall trapping to assess responses of the epigaeic invertebrate community to the ant suppression treatment. Four pitfall cups were placed in each plot, with one in the centre of each plot quarter. Pitfall traps consisted of a plastic cup of 69 mm diameter and 62 mm depth, filled with 50 ml of a 50:50 solution of water and propylene glycol. A small amount of unscented washing detergent was added to each 5-l container of trapping solution to reduce surface tension. We used PVC sleeves with a 69 mm inner diameter to ensure the same placement location of the trap in each sampling period and reduce the amount of soil disturbance when placing the cup in position. Lidded traps were placed 1 week before being filled with trapping solution, to reduce digging-in effects[55]. Traps were left open for 1 week before collection. Once collected, the samples were washed over a 125-µm sieve and stored in 70% ethanol at 5 °C until sorting. We selected pitfall traps because they sample surface-foraging invertebrates that are likely to be influenced by changes in ant abundance and composition in response to our treatments.

## Functional responses to suppression

We measured the performance rates of four ecological functions that are commonly associated with ants: scavenging for invertebrate carrion, myrmecochory (dispersal of elaiosome-bearing seed), granivory (seed predation) and plant defence against herbivores. For the three ground-resource-removal functions (scavenging, myrmecochory and granivory), we used transects of single-item resource placements a minimum of 30 cm apart, rather than placing resources in piles, to avoid high ant recruitment to baits and artificially inflated removal rates. We assessed rates of ant scavenging for invertebrate carrion as the proportion of dead mealworms (*Tenebrio molitor*; freshly killed, between 8 and 12 mm in length) at 30 min postplacement, whereas for myrmecochory we used the proportion of elaiosome-bearing seed (*Acacia acuminata*) removed at 60 min, and for granivory the proportion of seeds of two weed species common to the site (*Lolium rigidum* and *Erodium* sp., assessed at 60 min postplacement). Further detail for each function assessment can be found in Supplementary Note 4. We established 20 bait stations along a 6-m transect placed diagonally through each of two plot quarters in each plot for each resource. At each bait station (spaced 30 cm apart along each transect), a 15-mm wide and 5-mm deep depression was made in the soil and filled with a small quantity of white sand to act as a visual background contrasting with the darker-coloured food resources. The resource associated with the function being tested was then placed in the centre of each sand patch and assessed as 'removed' if not present within 15 cm of the original location after the predetermined period. Each resource being tested was set on its own transect within the plot quarter, and resources were assessed on different days. Whenever sampling was conducted, two sets of observations were carried out independently for each function, in each plot quarter, and the data were pooled at the plot-quarter level for analysis (that is, functional outcomes for 20 baits in each of two transects, repeated twice in each plot for each function, for a plot-level total of 80 baits per ground-resource-assessed functions per observation date). Each split-plot control–treatment pair within a given block was always run concurrently, but block-level observations were randomly stratified across consecutive days owing to logistical constraints.

Ants can be attracted to carbohydrates available in tree canopies, which may lead to protection of the plant through ant presence deterring herbivores. This effect may be driven by plants producing extrafloral nectar (EFN) or by ants forming mutualistic food-for-protection relationships with various honeydew-producing Hemiptera. As hempteran aggregations were variable across the site, and our selected plot types did not include EFN-bearing tree species, we used experimental 'surrogates' of extrafloral nectaries to enable more control of the availability of carbohydrates in trees. Experimental nectaries consisted of a 5-ml Eppendorf tube with a 3-mm hole drilled in the lid. This hole was plugged with a 1-cm piece of cotton wool, which was teased out a small distance. The experimental EFNs were attached to the tree with a cable-tie, with the EFN opening angled downwards, ensuring the cotton plug stayed moist and was within 3 mm of the tree bark. Experimental nectaries were loaded with a solution of water combined with 21.7% (wt./vol.) fructose, 18.9% sucrose and 4.1% glucose[56]. We placed eight experimental EFNs in each of two trees per plot quarter, ensuring that the trees were free of natural aggregations of Hemiptera, in the late afternoon on the day before the plant defence function was tested. We then tested whether ants foraging in these trees might deter chewing herbivores. On the day of the function assessment, we glued a single second instar mealworm larva (~7 mm long, dorsal side down) to the base of one young leaf at the apical growing tip and another mealworm to a mature leaf on each of four branches per tree. We monitored these larval baits for a total of 15 min and recorded them as 'attacked' if an aggressive interaction with an ant occurred. For the ground-resource assessments described above, we pooled observations at the plot level, with eight baits per tree and four trees per plot for a total of 32 baits.

## Calculation of multifunctionality

We calculated multifunctionality using a composite approach consisting of between-function covariance, effective multifunctionality[57] and a multiple-thresholds analysis[23]. This composite approach enabled thorough investigation of multifunctional performance and was required because no single measure adequately describes multifunctional rates, with trade-offs, positive and negative covariance, and between-function averaging potentially masking true variation in multifunctionality[21,58]. We defined the maximum observed value as the mean of the top 5% of observed functional rates across all observations. We report multiple-thresholds outputs as $T_{min}$ and $T_{max}$, indicating the minimum and maximum thresholds at which diversity significantly influences functioning, and $R_{mde}$, representing the slope at which the biodiversity–multifunctionality relationship is strongest[26,59].

## Statistical analyses

We performed mixed-effects modelling with the 'lme4' package[60] in R v.4.1.2 (ref. [61]) to test for an effect of ant suppression treatment on the remaining ant communities. Species richness ignores important variations in species commonness and rarity, which are also potentially important predictors of functioning, particularly in a context such as ours that is likely to involve treatment-driven changes in species dominance. Therefore, in addition to species richness, we used the ENS to determine whether observed biodiversity–multifunctionality relationships still held after accounting for effects of species relative abundance on functional rates[62,63]. To test for a signal of compositional change in response to treatment, we used a variant of the Raup–Crick dissimilarity metric[64] to calculate a measure of compositional 'uniqueness' of each community. More details regarding the calculation of ENS and the Raup–Crick dissimilarity metric can be found in Supplementary Note 5. We specified a block-level random effect to account for the split-plot design of the suppression and control treatments. We also used a plot-level random effect, nested within the block-level effect, to account for repeated measurements per plot. We specified a Poisson error distribution for abundance and richness responses and tested to ensure there was no overdispersion of model residuals, whereas for

ENS, community uniqueness, functional richness and functional dispersion, we specified Gaussian error distributions. We inspected residuals of Gaussian models for violations of normality and homoscedasticity.

Given that the shape of the SR–FR relationship is a key determinant of the presence of functional redundancy at higher species-richness values[37], we sought to investigate the shape of this relationship and whether it changed with treatment imposition. We used GAMMs accounting for non-independence of replicates in the experimental design (as detailed above) to test for a shift between linear and non-linear (asymptotic) SR–FR relationships, using the 'gamm' function in the 'mgcv' package[65]. An asymptotic relationship would tend to suggest high redundancy, with the addition (or loss) of a few species having little effect on functional richness, whereas a linear relationship would suggest high complementarity, with each species contributing a similar amount to functional richness. We used a model-comparison approach to determine the inclusion and significance of fixed effects in the best-fit GAMM, comparing five candidate models of increasing complexity: the null intercept-only model, a treatment-only model, a species-richness-only model, a model with different treatment-level intercepts but the same SR relationships, and a model with different treatment-level intercepts and different species-richness relationships for control versus suppression treatments. Candidate models were compared using the AIC with the 'AICcmodavg' package[66].

As our ant suppression treatment could have influenced function either directly through removal of individual foragers or indirectly through consequent changes in the richness or relative abundance of non-target ants, we used generalized multilevel confirmatory path analysis with the 'piecewiseSEM' R package[67] to tease apart the direct and indirect effects of our treatment manipulation on the performance of each function separately and on effective multifunctionality ($q = 1$). Unlike other approaches, this method of path analysis enables use of mixed-model approaches to account for random variance in spatial or temporal elements of an experimental design[67]. Component models were specified using lme4, with random effects of plot nested within experimental block, because our experiment had repeated measurements at the block level and a split-plot design in the manipulation. We repeated these analyses to test the potential indirect mediating effects of other community-level metrics (ENS and community composition) on functional responses.

In addition to the effective multifunctionality approach, we calculated the shift in biodiversity–multifunctionality relationships across multiple functional thresholds using the methods of Byrnes et al.[59] and Lefcheck et al.[23]. To calculate the biodiversity–multifunctionality relationship at each threshold, we used Poisson mixed models within lme4, with an observation-level random intercept added, as required, to account for overdispersion in model residuals. These models generated beta estimates (that is, standardized slopes of the relationship between species richness on the $x$ axis versus the number of functions simultaneously performed above a specified threshold level on the $y$ axis) that were plotted against each threshold level to determine the strength of the biodiversity–multifunctionality relationship in control versus ant suppression plots across multiple thresholds (that is, every 1% increment in threshold value between 5% and 95% of function). One of the difficulties with this method is that the analyses produce a curve with associated descriptive statistics rather than a single testable metric[59]. Therefore, to test whether our treatment had a statistically significant effect on multifunctional performance, we used lme4 to test for an effect of ant suppression on the number of functions performed at the single threshold representing the average $R_{mde}$ of our two multiple-threshold curves (these being the curves generated by multiple-thresholds analyses in each of our control and treatment zones). The multiple-thresholds analysis for species richness versus multifunctionality was repeated for an ENS versus multifunctionality relationship to test whether the observed relationships were strongly influenced by variations in species relative abundances.

We tested the relative associations of species with the performance of multiple functions and estimated their relative functional redundancy in response to suppression of dominant competitors, using the null-model approach of Gotelli et al.[68] to calculate a species-specific measure of association with function. Using measures of abundance of species captured in pitfall traps and rates of each function measured in the same plot, this method enables determination of associations between species occurrence and relative functional rates. These associations provide information on both direct and indirect relationships between species and functional performance. It was not logistically feasible to conduct direct field observations of functional performance at the individual species level for all combinations of function and suppression.

We compared regressions of species abundances against 999 randomizations of functional performance measures and calculated standardized effect sizes using the deviation of observed values from the average values generated by the null models. This analysis was conducted using the software 'Impact'[69]. The method uses randomization of functional rate (rather than species abundance), based on the assumption that species abundance influences functioning rather than functioning influencing species[68]. We ran species functional importance analyses separately for plots in the control and treatment zones for all single functions. We then subtracted the standardized effect sizes in the control plots from those of the treatment plots to generate a delta treatment value that reflected the change in species-specific functional importance between control and ant-suppression-treated plots. These delta values were ranked from smallest to largest, and we used Spearman's rho to test for an association between functional rate and size of the treatment effect.

### Reporting summary

Further information on research design is available in the Nature Portfolio Reporting Summary linked to this article.

## Data availability

All datasets used to produce the work described in this paper are available via figshare at https://doi.org/10.6084/m9.figshare.27998150 (ref. 70).

## Code availability

R code used to produce the work described in this paper is available via figshare at https://doi.org/10.6084/m9.figshare.27998150 (ref. 70).

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

## Acknowledgements

We acknowledge funding from the Australian Research Council via Discovery grant DP130102203 awarded to R.J.H. We thank M. Van Wees, H. White, B. Johnson and L. Mason for assistance in the field and laboratory and B. Heterick for confirmation of ant identifications. We thank those who planted and maintained the Ridgefield Tree Diversity experiment, in particular, R. Campbell, K. Hulvey, T. Morald, M. Perring and R. Standish.

## Author contributions

P.Y., R.K.D., R.J.H. and L.L. conceived the study and contributed to experimental design. P.Y. and R.K.D. performed the fieldwork and analysed the data. P.Y. drafted the paper, and all authors contributed to the final version.

## Funding

## Competing interests

The authors declare no competing interests.

## Additional information

**Extended data** is available for this paper at

**Supplementary information** The online version
contains supplementary material available at

**Correspondence and requests for materials** should be addressed to
Peter Yeeles or Raphael K. Didham.

**Peer review information** *Nature Ecology & Evolution*
thanks the anonymous reviewers for their contribution
to the peer review of this work. Peer reviewer reports are
available.

**Publisher's note** Springer Nature remains neutral with regard to
jurisdictional claims in published maps and institutional affiliations.

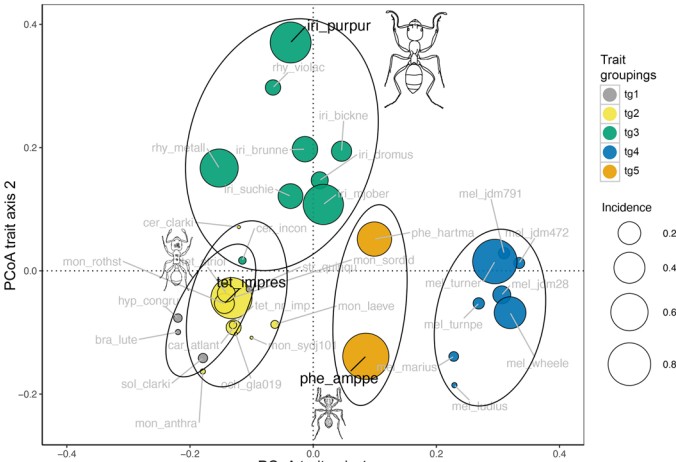

**Extended Data Fig. 1 | Trait-space representation for the naturally occurring ant community prior to suppression of dominant species.** Principal Coordinates Analysis (PCoA) ordination of ant species projected into trait space. The size of the bubbles is determined by site-wide incidence in 400 pitfall traps sampled in 2012. Ellipses are for illustrative purposes only, and colours represent functional group membership of each species as listed in the supplementary information. Species with names in bold text are those selected for experimental suppression (iri_pupur = *Iridomyrmex purpureus*, phe_amppe = *Pheidole ampla perthensis*, and tet_impres = *Tetramorium impressum*). Incidence data and a key to species code names are in supplementary information, Table S2.

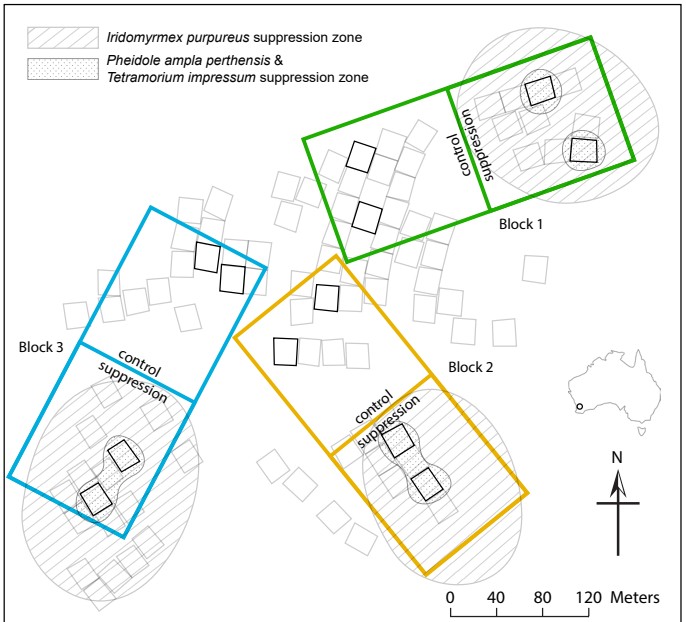

**Extended Data Fig. 2 | The layout of ant suppression treatments within the Ridgefield tree diversity experiment[36].** Coloured outlines show the positions of the three spatial blocks containing split plot control and suppression treatments. Black-outlined rectangles show the positions of the 12 focal plots used in this study, within the wider context of the 100 light-grey tree diversity experiment plots. Polygons filled with diagonal-line shading represent the approximate area of ant colony removals for suppression of *Iridomyrmex purpureus*, and polygons with stippled shading represent the areas of colony removals for suppression of *Tetramorium impressum* and *Pheidole ampla perthensis*. The inset map shows the location of study site within Australia.

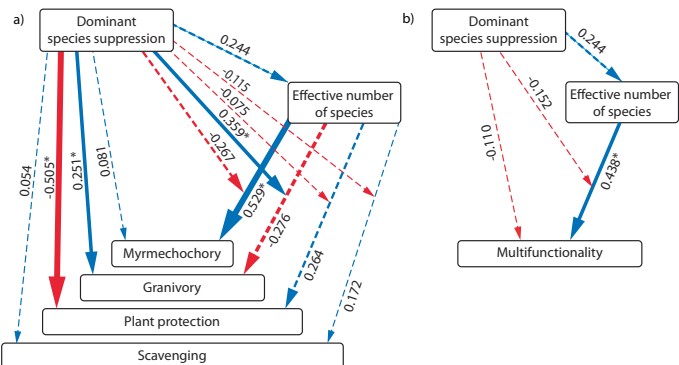

**Extended Data Fig. 3 | Results were qualitatively consistent when accounting for variation in species evenness using the effective number of species rather than species richness.** Path models showing the relationship between **a**) dominant species suppression and ecosystem function, as mediated by the effective number of species, and **b**) dominant species suppression and effective multifunctionality (q = 1), as mediated by the effective number of species. Species richness was calculated after removal of the target species from each dataset.

Arrow widths are scaled by the standardized partial regression coefficient (that is, effect size, as indicated by the values next to paths), coloured to represent the direction of effect (red arrows show negative effects, blue arrows show positive effects), and are solid rather than dashed for statistically significant effects (*, where bootstrapped 95% CI do not overlap zero). Note panel (a) in this figure is a composite of multiple SEM, each of which was estimated separately for each individual function.

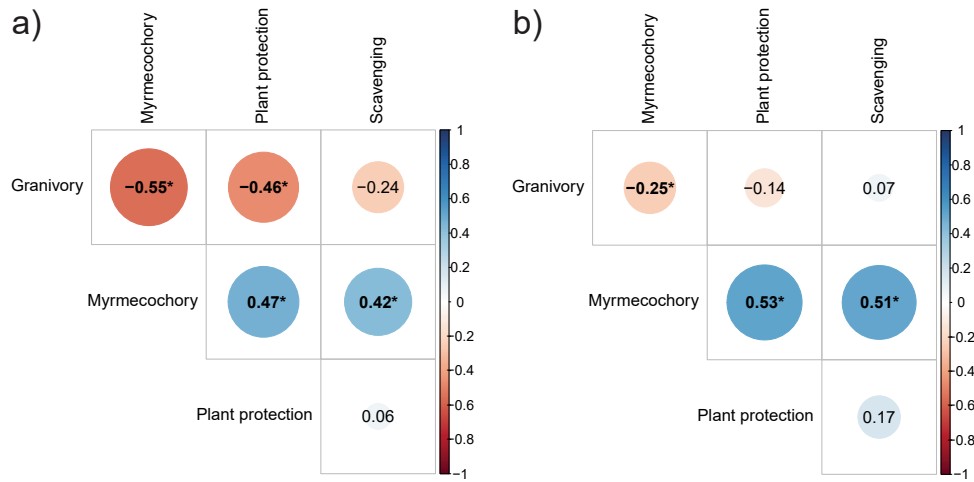

**Extended Data Fig. 4 | The sign and magnitude of plot-level covariance among functions varied following dominant species suppression.** Standardised correlations are shown among rates of the four measured functions in **a**) control plots and **b**) ant suppression plots. Significant correlations highlighted in bold.

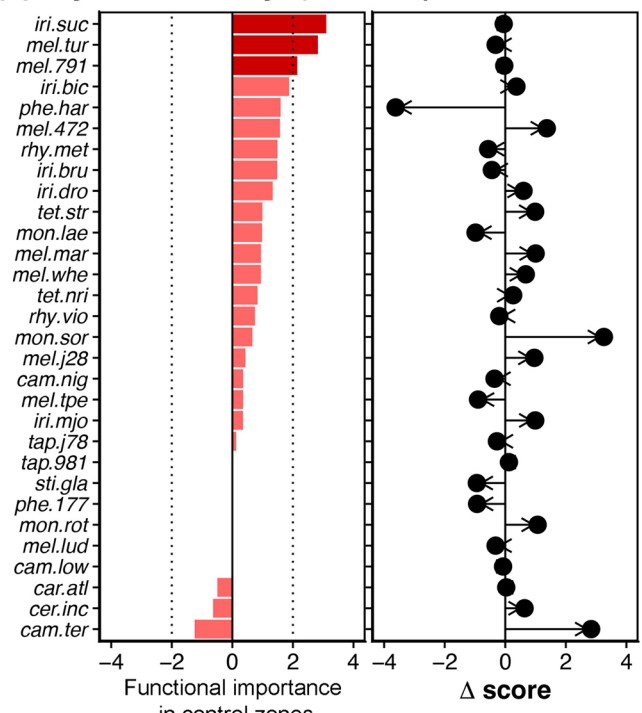

**(a) Myrmecochory:** ρ -0.11, p = 0.56

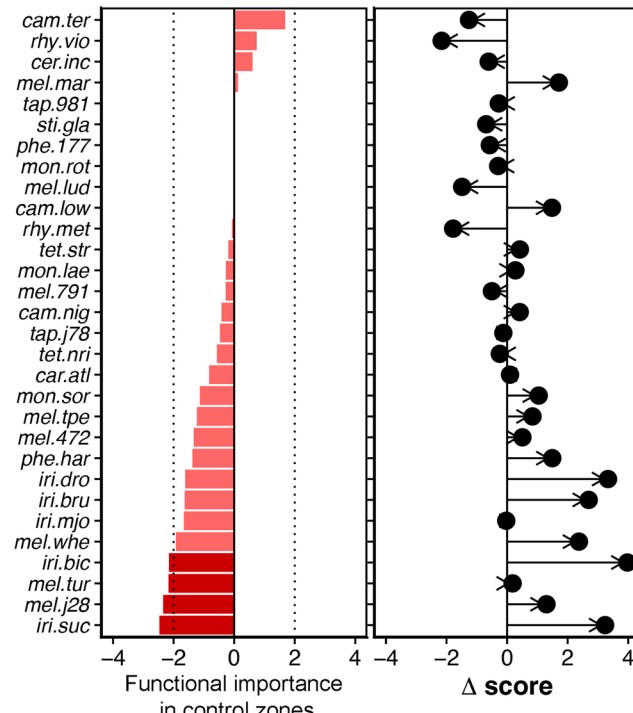

**(b) Granivory:** ρ -0.69, p < 0.05

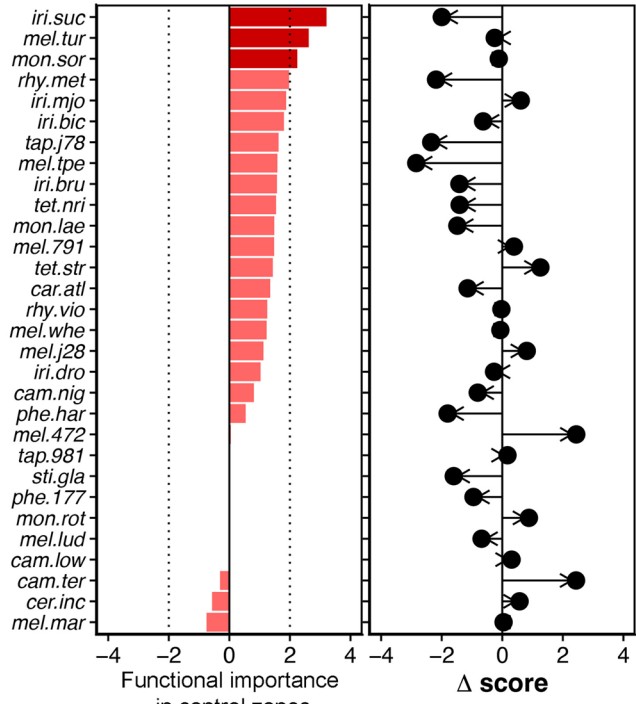

**(c) Plant protection:** ρ -0.42, p < 0.05

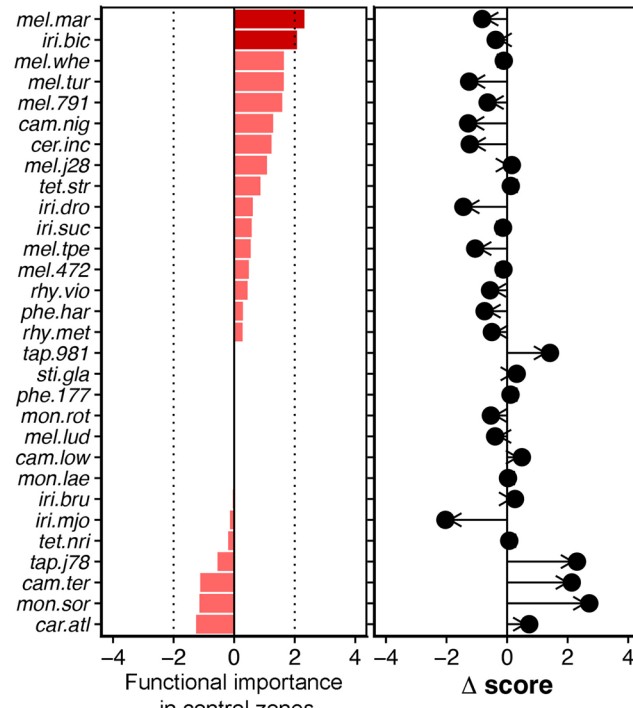

**(d) Scavenging:** ρ -0.58, p < 0.05

**Extended Data Fig. 5 | Species with the greatest increase in functional contribution in the suppression plots tended to be the ones most negatively associated with functional rates in the control plots in the presence of dominant competitors.** A comparison of changes in species functional importance scores between control plots and the dominant ant suppression treatment for (**a**) myrmecochory, (**b**) granivory, (**c**) plant protection and (**d**) scavenging. The delta (Δ) score represents the relative shift in functional importance in the control zones versus the suppression zones (positive values indicate that a species had a higher degree of association with functional rates following the ant suppression treatment, while negative values indicate a decreased association). Effects shaded in darker red indicate significant delta scores that fall outside the 95% CI of a normal distribution of scores generated by a null model approach. Treatment-level variation species abundances and a key to species code names are in supplementary information, Table S3.

Raphael K Didham

# Reporting Summary

## Statistics

For all statistical analyses, confirm that the following items are present in the figure legend, table legend, main text, or Methods section.

| n/a | Confirmed | |
|---|---|---|
| ☐ | ☒ | The exact sample size (*n*) for each experimental group/condition, given as a discrete number and unit of measurement |
| ☐ | ☒ | A statement on whether measurements were taken from distinct samples or whether the same sample was measured repeatedly |
| ☐ | ☒ | The statistical test(s) used AND whether they are one- or two-sided *Only common tests should be described solely by name; describe more complex techniques in the Methods section.* |
| ☐ | ☒ | A description of all covariates tested |
| ☐ | ☒ | A description of any assumptions or corrections, such as tests of normality and adjustment for multiple comparisons |
| ☐ | ☒ | A full description of the statistical parameters including central tendency (e.g. means) or other basic estimates (e.g. regression coefficient) AND variation (e.g. standard deviation) or associated estimates of uncertainty (e.g. confidence intervals) |
| ☐ | ☒ | For null hypothesis testing, the test statistic (e.g. *F*, *t*, *r*) with confidence intervals, effect sizes, degrees of freedom and *P* value noted *Give P values as exact values whenever suitable.* |
| ☒ | ☐ | For Bayesian analysis, information on the choice of priors and Markov chain Monte Carlo settings |
| ☐ | ☒ | For hierarchical and complex designs, identification of the appropriate level for tests and full reporting of outcomes |
| ☐ | ☒ | Estimates of effect sizes (e.g. Cohen's *d*, Pearson's *r*), indicating how they were calculated |

*Our web collection on statistics for biologists contains articles on many of the points above.*

## Software and code

Policy information about availability of computer code

| Data collection | No software was used for data collection. |
|---|---|
| Data analysis | All analyses were performed with the open source R (version 4.1.2).<br><br>We used the 'lme4' package to test for an effect of ant suppression treatment and the 'piecewiseSEM' package for piecewise structural equation models. We used the software 'Impact' (version 1.0) to compare species abundances with functional performance measures, and 'multifunc' for analysing functional performance data. Plots were generated using the ggplot2 R package.<br><br>Code is available to download from Figshare: https://doi.org/10.6084/m9.figshare.27998150 (this DOI is referenced in the manuscript). |

For manuscripts utilizing custom algorithms or software that are central to the research but not yet described in published literature, software must be made available to editors and reviewers. We strongly encourage code deposition in a community repository (e.g. GitHub). See the Nature Portfolio guidelines for submitting code & software for further information.

## Data

Policy information about availability of data

All manuscripts must include a data availability statement. This statement should provide the following information, where applicable:
- Accession codes, unique identifiers, or web links for publicly available datasets
- A description of any restrictions on data availability
- For clinical datasets or third party data, please ensure that the statement adheres to our policy

Raw data are available to download from Figshare: https://doi.org/10.6084/m9.figshare.27998150 (this DOI is referenced in the manuscript).

# Field-specific reporting

Please select the one below that is the best fit for your research. If you are not sure, read the appropriate sections before making your selection.

☐ Life sciences ☐ Behavioural & social sciences ☒ Ecological, evolutionary & environmental sciences

For a reference copy of the document with all sections, see nature.com/documents/nr-reporting-summary-flat.pdf

# Ecological, evolutionary & environmental sciences study design

All studies must disclose on these points even when the disclosure is negative.

| | |
|---|---|
| Study description | This study uses an experimental manipulation of ant species diversity to investigate the relative importance of functional redundancy versus complementarity in conferring the stability of multi-functional ecosystem function performance. We suppressed the abundance of dominant ant species from multiple trait groupings and measured subsequent changes in the rates of 4 important ecosystem processes. |
| Research sample | Our research samples consisted of ant specimens and records of the ecological processes they performed. |
| Sampling strategy | We used 4 pitfall traps per plot to assess invertebrate community responses to the ant suppression treatment. Pitfall cups were 69mm diameter and 62mm in depth, filled with 50 ml of a 50:50 water and propylene glycol preservative. Traps were left open for one week durations. |
| Data collection | Observations were carried out only in calm and sunny weather, from 8:30 AM to 5:00 PM. |
| Timing and spatial scale | Sampling and observations took place every three months for one year from treatment imposition, in November 2014, and February, May and November 2015. |
| Data exclusions | No data were excluded from the analysis. |
| Reproducibility | All data necessary to repeat the analyses will be made publicly available. |
| Randomization | The site layout within the context of a broader ecosystem function experiment combined with method of treatment application did not allow randomization of site-treatment allocation. |
| Blinding | Information regarding treatment level was blinded during taxonomic identification of insect voucher specimens. |

Did the study involve field work? ☒ Yes ☐ No

## Field work, collection and transport

| | |
|---|---|
| Field conditions | The main habitat type is restoring Eucaluptus forest. The climate is Mediterranean, with hot dry summers and cool wet winters. Median annual precipitation in this area is 443 mm, mainly concentrated in winter between April and September The average annual temperature ranges from 10.4 °C mean minimum to 23.5 mean maximum °C. |
| Location | The study was carried out in at the Ridgefield Multiple Ecosystem Services experiment, located near Pingelly in Western Australia, Australia. |
| Access & import/export | No permits were required to perform this work. |
| Disturbance | No disturbance was caused in the study sites. |

# Reporting for specific materials, systems and methods

We require information from authors about some types of materials, experimental systems and methods used in many studies. Here, indicate whether each material, system or method listed is relevant to your study. If you are not sure if a list item applies to your research, read the appropriate section before selecting a response.

## Materials & experimental systems

| n/a | Involved in the study |
|---|---|
| ☒ | Antibodies |
| ☒ | Eukaryotic cell lines |
| ☒ | Palaeontology and archaeology |
| ☐ ☒ | Animals and other organisms |
| ☒ | Human research participants |
| ☒ | Clinical data |
| ☒ | Dual use research of concern |

## Methods

| n/a | Involved in the study |
|---|---|
| ☒ | ChIP-seq |
| ☒ | Flow cytometry |
| ☒ | MRI-based neuroimaging |

## Animals and other organisms

Policy information about studies involving animals; ARRIVE guidelines recommended for reporting animal research

| | |
|---|---|
| Laboratory animals | *For laboratory animals, report species, strain, sex and age OR state that the study did not involve laboratory animals.* |
| Wild animals | This study involved sampling wild ants. Ants were collected using pitfall traps filled with liquid preservative, which killed ants upon capture. A list of species captured is included in Supplementary Table S2. Ants were preserved either dry, on card points, or wet in ethanol and identified in a laboratory after fieldwork. Curated specimens were lodged at the West Australian Museum. |
| Field-collected samples | *For laboratory work with field-collected samples, describe all relevant parameters such as housing, maintenance, temperature, photoperiod and end-of-experiment protocol OR state that the study did not involve samples collected from the field.* |
| Ethics oversight | As this project only involved the sampling of ants, there was no ethics oversight required under Australian law. |

Note that full information on the approval of the study protocol must also be provided in the manuscript.

