## [Peer Review File · Nature Ecology & Evolution]

Functional redundancy compensates for decline of dominant ant species

Corresponding Author: Professor Raphael Didham

Version 0:

Decision Letter:

Dear Peter,

Thank you for your patience while your Article entitled "Functional redundancy compensates for the decline of dominant insect species" was under review. We have now received reports from two reviewers. As you will see from their comments copied below, they find your work of considerable potential interest but Reviewer 1 has raised important concerns that cast doubt on the strength of the conclusions. In light of these concerns, we will need to see a revised version that addresses the reviewers' concerns to be able to make a decision on your manuscript.

We hope you will find the reviewers' comments useful as you decide how to proceed. Importantly, as Reviewer 1's central criticisms are on the experimental design and the statistical analysis, it will be essential not only to clarify the description of the methods (and to carry out additional analyses if feasible), but also to provide the raw data and the R code. We also agree with the referee that a schematic of the experimental design would be useful.

* Highlight all changes in the manuscript text file and provide a version in Microsoft Word format.

* Include a "Response to reviewers" document detailing, point-by-point, how you addressed each referee comment. If no action was taken to address a point, you must provide a compelling argument. This response will be sent back to the referees along with the revised manuscript.

* Include a Data Availability and a Code Availability statement and provide the data and R script either in supplementary files or in a repository which the reviewers can access without waiving their anonymity.

* If you have not done so already, we suggest that you begin to revise your manuscript so that it conforms to our Article format instructions at <http://www.nature.com/natecolevol/info/final-submission>. Refer also to any guidelines provided in this letter.

Once ready, please use the link below to submit a revised paper:

Link Redacted

We would hope to receive a revised manuscript within 3 months. If you cannot send it within this time, please let us know. We will be happy to consider your revision so long as nothing similar has been accepted for publication at Nature Ecology & Evolution or published elsewhere.

Nature Ecology & Evolution is committed to improving transparency in authorship. As part of our efforts in this direction, we are now requesting that all authors identified as 'corresponding author' on published papers create and link their Open Researcher and Contributor Identifier (ORCID) with their account on the Manuscript Tracking System (MTS), prior to acceptance. This applies to primary research papers only. ORCID helps the scientific community achieve unambiguous attribution of all scholarly contributions. You can create and link your ORCID from the home page of the MTS by clicking on 'Modify my Springer Nature account'. For more information please visit www.springernature.com/orcid.

Thank you for the opportunity to review your work.

[redacted]

Reviewers' comments:

Reviewer #1 (Remarks to the Author):

General comments

The manuscript presents a novel test of the importance of functional redundancy for multifunctional performance. Experimental work in this area of research usually considers plants as plant communities are relatively easy to manipulate. This study is significant in that it uses species removal experiments in a diverse, reassembled assemblage of animals (ants) and the experiments convincingly reduce abundances of the dominant species by up to 99%. Such experiments are very difficult to conduct effectively and results should be of immediate interest to people interested in ecosystems at all levels.

The biggest flaw I can see with this experiment is that it is, as described, a split plot design with three replicate plots, but the analyses ignore this design, which brings the results and conclusions into question. It's always helpful to see a map to make it very clear what the design is. As I understand it, the two replicates within each of the three plots are not really replicates, but subsamples. Further, in the plots, it looks like pitfall traps are treated as replicates, when normally a set of pitfall traps within a plot/site would be pooled for a site/plot total. Thus, my impression is that the number of independent replicates is vastly inflated in the analyses. I understand that these are very difficult experiments to conduct, but that doesn't justify this treatment of the data. I think there could be other ways of treating this data that might be valid, for example, for Fig. 1, the functional richness curve could be presented for each of the three replicates and the analysis could account for the random effect of site. If this analysis shows similar patterns to those currently presented, then it could be convincing, but I'm not sure it would be powerful enough to show this statistically with just three replicates.

I consider the conclusions to be original for an animal group – I'm not aware of other studies that have broken down animal communities in this way. However, the issues with replication that I mentioned above mean that I'm not convinced of the conclusions.

Minor comments

A minor issue is that Delta AICc values are presented in lines 101 and 102, but the second of these is very small, suggesting that the linear model was not meaningfully better than other models. I couldn't find a table of AICcs in the appendix to check whether my interpretation was correct.

Strangely, in figure 2, 84% confidence intervals are given. I couldn't find an explanation for this unusual value for confidence intervals.

It wasn't clear if all three species needed to be removed from each block.

Observation of actual use of food resources would have provided more reliable data than Gotelli's method of estimation. For example, it seems strange that *Iridomyrmex* turns up under granivory. Visual observations could confirm this.

The Pilot study didn't really test the requirements for sample size, but was more a test of whether the treatment works.

I struggled to understand the threshold analysis, so it would be helpful to provide a fuller explanation (rather than just referring to the Byrnes and Lefcheck papers).

Some comments were left on the extended data files

Iridomyrmex purpureus appears to be missing from Table S3. Also, from Table 3, justification of these being dominant species is unclear: the abundance of the "dominant" species *Iridomyrmex purpureus* (missing, but plausibly dominant), *Pheidole ampla perthensis* (data missing), *Tetramorium impressum* (3 in control and 11 in suppression plots), *Monomorium sordidum* (218, 240 – plausibly dominant), and, *Melophorus turneri* (425, 749, plausibly dominant).

Reviewer #2 (Remarks to the Author):

In this study, the authors use experimental removal of dominant ant species to test if the remaining species compensate the loss of ecological functions through functional redundancy effects. This question is both timely and touches on fundamental theory within the biodiversity-ecosystem functioning framework and therefore highly relevant. The authors manage to answer this theoretical question for their study system using an approach which is both elegantly designed and thoroughly planned

and conducted. In addition, the authors did a very good job in documenting their approach, making their reasoning and decision process clear and in describing everything in a clear way. It was really a pleasure to read such a high quality manuscript.

I do have two suggestions for further improvement:

1. The abstract is a bit too abstract and technical in the results part. While it has to be concise and this is difficult with a theoretical background and counter-intuitive results, I think readers would benefit from simpler sentences and less detail. For example, the information on how much the trait dispersion was reduced is not necessary here. The sentence in line 21 is also very long and complicated.

2. The fact that functional performance was measured by actually looking at ant activity around baits (and was not only inferred through the abundance of certain traits within pitfall traps for example) is only mentioned "in passing" during the first part of the manuscript. Only when the reader arrives at the figure on granivory etc. does it become clear that functions were observed in the field. I would recommend to mention this part of the study explicitly early in the manuscript as this is a really strong addition to the study overall.

Minor comments

Line 108: There is a full stop missing at the end of the sentence

Line 120: Confidence areas are shown by shaded areas not by dashed lines

Version 1:

Decision Letter:

4th February 2025

Dear Dr. Yeeles,

Thank you for your patience while your revised manuscript entitled "Functional redundancy compensates for the decline of dominant insect species" was under review. In light of the reviewers' advice, we will be happy in principle to publish it in Nature Ecology & Evolution, pending minor revisions to satisfy the reviewers' final requests and to comply with our editorial and formatting guidelines.

[redacted]

Reviewer #1 (Remarks to the Author):

The authors have done an excellent job of revising analyses and improving clarity on how the analyses account for the subsamples. Their response addresses all my issues with that aspect of the manuscript.

The minor issues have also largely been addressed. It still concerns me that the functional importance of species figures (extended data figure 5) suggest that *Iridomyrmex* are important contributors to granivory. This suggests that the approach is flawed if it regards the species as performing the functions attributed to them via Gotelli's method. As the authors suggest, the apparent association between *Iridomyrmex* and granivory might be due to interactions with other species that are granivores. If this is actually a measure of the association of each species with the community-wide function, rather than the specific functional contribution of that species (via its diet, and depending on the co-occurring species), then that should be clarified in the manuscript.

Reviewer #2 (Remarks to the Author):

The first version of the manuscript has already been of very high quality and my rather minor comments have been addressed very well.

Reviewer #1 (Remarks to the Author):

General comments

The manuscript presents a novel test of the importance of functional redundancy for multifunctional performance. Experimental work in this area of research usually considers plants as plant communities are relatively easy to manipulate. This study is significant in that it uses species removal experiments in a diverse, reassembled assemblage of animals (ants) and the experiments convincingly reduce abundances of the dominant species by up to 99%. Such experiments are very difficult to conduct effectively and results should be of immediate interest to people interested in ecosystems at all levels.

Response 1.1: We thank the reviewer for the positive feedback.

The biggest flaw I can see with this experiment is that it is, as described, a split plot design with three replicate plots, but the analyses ignore this design, which brings the results and conclusions into question. It's always helpful to see a map to make it very clear what the design is. As I understand it, the two replicates within each of the three plots are not really replicates, but subsamples. Further, in the plots, it looks like pitfall traps are treated as replicates, when normally a set of pitfall traps within a plot/site would be pooled for a site/plot total. Thus, my impression is that the number of independent replicates is vastly inflated in the analyses. I understand that these are very difficult experiments to conduct, but that doesn't justify this treatment of the data. I think there could be other ways of treating this data that might be valid, for example, for Fig. 1, the functional richness curve could be presented for each of the three replicates and the analysis could account for the random effect of site. If this analysis shows similar patterns to those currently presented, then it could be convincing, but I'm not sure it would be powerful enough to show this statistically with just three replicates.

Response 1.2: We thank the reviewer for this important and insightful comment.

We address this issue in detail below, with two general responses: (1) we obviously failed to explain the analyses in a clear manner, because the majority do already take into account the non-independence of individual replicates in the split plot design; and (2) the reviewer is correct that the analysis in Figure 1 is wrong, and in hindsight we should have used a mixed effects modelling framework here as we did in all other analyses.

(1) First, we confirm that the reviewer's description of the experimental design is correct (3 spatial blocks, within which there was a split-plot allocation of control versus suppression zones, each containing two nested plots sampled multiple times).

As requested, we have now added a schematic map of the experimental layout as Extended Data Figure 2 and we cite the figure at line 77 and line 428. As we describe in the manuscript, one key advantage of our experiment is that it is located in the 'common-garden' setting of the 'Ridgefield' TreeDivNet experimental tree diversity site in

southwestern Australia, and benefits from highly controlled components such as tree species diversity treatments and planting layout. We sampled ant communities for 4 years prior to the ant suppression experiment, which allowed us to measure underlying variation in community composition, and identify the need for spatial blocking of the ‘Ridgefield’ site. As described in the manuscript, the ‘suppression’ zone in each block was paired with an adjacent ‘control’ zone in the nearest suitable plots that had the same standardised tree-planting treatment (see Extended Data Figure 2).

We definitely recognise the non-independence of replicates in this split plot design (i.e. each data point within individual plots is a pseudo-replicate not a true independent replicate). Although we freely acknowledge that we made an error in the analysis depicted in Figure 1 (which we fully address below), we would like to point out that all of our remaining analyses did already take the split plot design into account using a mixed effects modelling framework with random effects for ‘spatial block’ and ‘plot nested within block’.

For example, at L.578 in the Methods section we write: *“We used a mixed effect modelling approach performed with the ‘lme4’ package in R version 4.1.2 to test for an effect of ant suppression treatment on the remaining ant communities.”* and *“We specified a block level random effect to account for the split plot design of the suppression and control treatments. We also used a plot level random effect, nested within the block level effect, to account for repeated measurements per plot.”*

Similarly, at L. 612 we write: *“Unlike other approaches, this method of path analysis allows the use of mixed model approaches to account for random variance in spatial or temporal elements of an experimental design. Component models were specified using ‘lme4’ with random effects of plot nested within experimental block, because our experiment had repeated measurements at the block level, and a split-plot design in the manipulation.”*

Nevertheless, we obviously need to improve the clarity of explanation of these points in the manuscript to avoid confusion. We now add additional text in the main manuscript, at L.91, explaining that ant suppression effects are tested *“after accounting for non-independence of replicates in the split plot experimental design using a mixed effects modelling framework”*, and similar statements are added to the captions of Figure 1 and Figure 2.

(2) Notwithstanding our ‘intent’ to fully incorporate non-independence of replicates in all analyses (as detailed above), we did make an unintentional error in the analysis of the species richness versus functional richness (SR-FR) relationship in Figure 1. In hindsight this seems like an obvious mistake (and we are not quite sure how this happened), but we now correct this by re-analysing the SR-FR relationship in a mixed-effects modelling framework.

Given the non-linear (putatively asymptotic) relationship being tested, standard ‘linear’ mixed models are not necessarily appropriate for these data. There are two common alternative statistical approaches to this problem, (i) generalized additive mixed models (GAMMs to fit non-linear smooths) and (ii) segmented mixed models (to fit ‘breakpoint’ changes in the slope of linear trends), and we carry out both of these to ensure that the outcome is robust to alternative statistical approaches. We find that both of these approaches produce the same conclusions as identified earlier from the incorrect SSasymp

approach. Given that the conclusions are consistent, we currently only include the GAMM approach in the revised manuscript, but if the reviewer or Editor prefer, we are open to including the segmented mixed model results as well (perhaps in the Supplementary Information).

(i) generalized additive mixed models

We fitted GAMM models to the SR-FR relationship in Figure 1 using the ‘mgcv’ package in R (as described in more detail in Supplementary Information). This approach explicitly accounts for the non-independence among samples using random effects for ‘spatial block’ and ‘plot nested within block’, as used in all other analyses.

We used a model comparison approach to determine the inclusion and significance of fixed effects in the best-fit GAMM, comparing five candidate models of increasing complexity: the null intercept-only model, a treatment-only model, a species richness(SR)-only model, a model with different treatment-level intercepts but the same SR relationships, and a model with different treatment-level intercepts and different SR relationships for control versus suppression treatments. The best-fit GAMM was the full model with differing SR-FR relationships for control vs suppression treatments, after accounting for block and plot level random effects:

```
gamm.mod.txr <- gamm(FRic ~ treatment +
  s(richness.long.cs, by=as.factor(treatment), bs="fs"),
  random=list(exp.block=~1, plot=~1),
  data=dat,
  family=gaussian(link=identity),
  gamma=1.4,
  select=TRUE,
  niterPQL=100,
  control=list(opt="optim", maxIter = 10000, msMaxIter=10000))
```

Model	Name	k	LogLik	AIC	ΔAIC	AIC weight	R-squared
5	treatment * SR(smooth)	9	224.21	-428.14	0.00	0.87	0.66
4	treatment + SR(smooth)	7	219.89	-424.39	3.75	0.13	0.63
3	SR(smooth) only	6	212.21	-411.40	16.74	0.00	0.51
2	treatment only	5	181.02	-351.32	76.83	0.00	0.05
1	intercept only (null)	4	178.53	-348.58	79.56	0.00	0.00

We plotted the predicted GAMM relationships from the best-fit model, and used different symbol shapes to denote observed values from the three spatial blocks.

It is evident that the fitted relationship is almost identical to the earlier SSasympt approach; i.e. a plateau in the SR-FR relationship for control plots and an approximately linear SR-FR relationship for suppression plots, even after accounting for the random effects in the model.

In GAMM models there are no specific ‘tests’ of the asymptotic nature of fitted smooths, but the 95% confidence intervals around model predictions can be used to gauge the clear differences between treatments (see Supplementary Information for further details).

(ii) segmented mixed models

To ensure that the conclusions of the SR-FR analysis are robust to the type of analytical approach selected to deal with non-linear relationships, we re-ran the analysis using segmented mixed models (SMM) in package ‘segmented’ in R, and present those results here.

Segmented models take a fitted linear mixed effects model and test whether piecewise segmentation of a single linear relationship into two (or more) separate linear components around a threshold ‘breakpoint’ improves model fit.

As in the GAMM analysis, we used a model comparison approach to determine the inclusion and significance of fixed effects in the best-fit segmented model, comparing nine candidate models of increasing complexity: five LMM including the null intercept-only model, a treatment-only model, a species richness (SR)-only model, a model with different treatment-level intercepts but the same SR slopes, and a model with different treatment-level intercepts and different SR slopes; plus four segmented mixed models (SMM), including a segmented SR-only model, a model with different treatment-level intercepts but the same segmented SR slopes, a model with different treatment-level intercepts and different segmented SR slopes but the same breakpoints for each treatment, and finally a model with different treatment-level intercepts, different segmented SR slopes and different breakpoints for each treatment. Of these nine models, two had equivalent fit in AIC comparisons (i.e. delta AIC within 2 units of one another), and the most parsimonious of these (i.e. with fewest fitted parameters, k) was the model with different segmented relationships for control versus suppression plots, but a single common breakpoint.

```
lme.txr <- lme(FRi ~ richness.long.cs * treatment,
  random = ~1 | exp.block/plot,
  na.action = na.omit,
  method = "ML",
  data = dat)
fit.seg.txr2 <- segmented(lme.txr, ~richness.long.cs, x.diff = ~treatment,
  random = list(exp.block = pdDiag(~1 + richness.long.cs)),
```

```
control=seg.control(it.max=15, n.boot=10, display=TRUE))
```

Model	Analysis	Model name	k	LogLik	AIC	Δ AIC	AIC weight
8	SMM	treatment * SR(seg; same.break)	10	230.80	-441.60	0.00	0.63
9	SMM	treatment * SR(seg; diff.breaks)	11	231.23	-440.46	1.13	0.36
7	SMM	treatment + SR(seg)	8	224.24	-432.48	9.12	0.01
4	LMM	treatment + SR	6	216.28	-419.53	22.07	0.00
5	LMM	treatment * SR	7	216.29	-417.20	24.40	0.00
6	SMM	SR(seg) only	7	213.14	-412.28	29.32	0.00
3	LMM	SR only	5	209.85	-408.98	32.61	0.00
2	LMM	treatment only	5	181.02	-351.32	90.28	0.00
1	LMM	intercept only (null)	4	178.53	-348.58	93.01	0.00

Again, the conclusions are qualitatively the same in the segmented mixed model analysis, with a plateau in the SR-FR relationship for control plots, and an approximately linear SR-FR relationship for suppression plots, even after accounting for the random effects in the model.

Segmented models do allow a specific test of whether SR-FR relationship reaches an asymptote about the threshold breakpoint (i.e. whether the 95% CL around the fitted slope include zero). In this case, all the fitted slopes are significantly different from zero, although the negative slope for control plots above the breakpoint is only marginally below zero.

```
slope(fit.seg.txr2, by=list(treatment="control")) #baseline level
slope(fit.seg.txr2, by=list(treatment="treatment"))
```

Treatment	segment	Slope	SE	t	0.95.low	0.95.up
control	leftSlope	0.086	0.013	6.780	0.061	0.111
control	rightSlope	-0.038	0.018	-2.124	-0.074	-0.003
suppression	leftSlope	0.066	0.013	4.911	0.039	0.092
suppression	rightSlope	0.032	0.012	2.633	0.008	0.056

I consider the conclusions to be original for an animal group – I’m not aware of other studies that have broken down animal communities in this way. However, the issues with replication that I mentioned above mean that I’m not convinced of the conclusions.

Response 1.3: Thank you once again for the positive comments on the importance and generality of the work. We have made every effort to address the central issue regarding ‘pseudoreplication’ above. All our analyses now include appropriate random effects structures to account for the non-independence of multiple pitfall samples collected within plots, multiple plots within treatments, and the split-plot design of treatment contrasts paired within the three spatial blocks. All our conclusions remain unchanged by the re-analysis of the data in Figure 1, therefore we feel confident in the robustness and generality of the findings.

Minor comments

A minor issue is that Delta AICc values are presented in lines 101 and 102, but the second of these is very small, suggesting that the linear model was not meaningfully better than other models. I couldn’t find a table of AICcs in the appendix to check whether my interpretation was correct.

Response 1.4: This is a good point. We now provide clear AIC comparisons of competing models for the new GAMM and segmented model analyses in the tables above, and present the GAMM AIC values in Supplementary Table S4. The table shows that the difference between the model with the lowest AIC and the next best model is 3.75. A delta AICc of >2 is considered to indicate a difference in model fit.

Strangely, in figure 2, 84% confidence intervals are given. I couldn’t find an explanation for this unusual value for confidence intervals.

Response 1.5: The 84 % confidence interval is used purely as a visual aid for readers, and not as any form of critical threshold in a statistical test. The concept, as outlined in the references below, is that two non-overlapping 84% CI are a ‘visual’ indication of means that are significantly different at the $p = 0.05$ level, whereas two means with non-overlapping 95% CI actually differ significantly at the $p = 0.01$ level. We reiterate that this does not influence the statistical tests performed, it is just a visual aid.

MacGregor-Fors I, Payton ME (2013) Contrasting Diversity Values: Statistical Inferences Based on Overlapping Confidence Intervals. PLoS ONE 8(2): e56794. <https://doi.org/10.1371/journal.pone.0056794>

Harvey Goldstein; Michael J. R. Healy. The Graphical Presentation of a Collection of Means, *Journal of the Royal Statistical Society*, Vol. 158, No. 1. (1995), p. 175-177.

Cumming, Geoff; Finch, Sue. Inference by Eye: Confidence Intervals and How to Read Pictures of Data, *American Psychologist*, Vol 60(2), Feb-Mar 2005, p. 170-180.

We have added an explanation for the use of 84% confidence intervals to the caption to Figure 2 (line X) and now provide further justification to the Supplementary Information. We re-plot Figure 2 here (below) using 95% confidence intervals for comparison. We would be happy to replace Figure 2 in the manuscript with this version, if preferred by the reviewer or Editor.

It wasn't clear if all three species needed to be removed from each block.

Response 1.6: We have added text on line 466 to clarify that all three species were targeted for removal from all suppression treatment zones in all three blocks.

Observation of actual use of food resources would have provided more reliable data than Gotelli's method of estimation. For example, it seems strange that *Iridomyrmex* turns up under granivory. Visual observations could confirm this.

Response 1.7: We agree that species-level performance data for the four measured functions would have been ideal to record in the field. Unfortunately, we did not have the available time or financial resources available to carry out species-specific observations at all baits at all locations. To our knowledge, this has never been conducted for all species in a community-level study before.

Although we do not discount the importance of direct visual observations (when available), we do see complementary advantages in using a null model approach to calculate a species-specific measure of contribution to function. Principal amongst these is that it could provide insight into possible indirect effects of species contribution to functional performance. For example, we agree that *Iridomyrmex* would not typically be considered granivorous, but the increasing relative contribution to functional performance following dominant species removal could stem from *Iridomyrmex* indirectly benefitting granivorous species (perhaps through competitive suppression of a less-effective granivore, for example).

Although these types of effects are inherently speculative, they do highlight important correlative associations that would be key targets for future experimental manipulation studies. Ultimately, these types of cause-and-effect relationships can only be teased apart by experimental removals (or additions) of species, as we conduct here, but it would be challenging logistically to conduct all possible combinations of species removals at the present time. Consequently, we feel there is a valuable purpose in using Gotelli's method of estimation.

The Pilot study didn't really test the requirements for sample size, but was more a test of whether the treatment works.

Response 1.8: Yes, that is correct. We had some initial concerns that nests of some of the five species selected for suppression could be challenging to locate and destroy, so we conducted a small-scale trial to assess the feasibility of our methodology. We now replace the term 'pilot study' with 'methodological trial' throughout the manuscript and supplementary text.

I struggled to understand the threshold analysis, so it would be helpful to provide a fuller explanation (rather than just referring to the Byrnes and Lefcheck papers).

Response 1.9: We agree that the multiple thresholds analysis is challenging to interpret. Because of this, our approach was to have a relatively short technical reference to the Byrnes methodology paper in our Methods section, but incorporate a more complete

explanation in ‘plain language’ in the main body of the manuscript as well as in the caption to Figure 4.

We have revisited all three of those sections to try and improve clarity even further, and we have added additional sentences to provide a fuller explanation:

Main text: *“The threshold analysis evaluates whether diversity becomes increasingly important in the simultaneous performance of multiple functions above a specified threshold level. In multiple thresholds analyses, diversity always had a positive effect on multifunctionality (i.e., ‘Diversity effect’ values above zero in Figure 4), but the BD-MF effect was weaker in control plots where negative covariance among functions was higher, and stronger in the ant suppression plots.”* L.208.

Caption to Figure 4: *“Diversity effect values above 0.0 represent positive BD-MF relationships.”*

Methods section: *“These models generated beta estimates (i.e. standardized slopes of the relationship between species richness on the x-axis versus number of functions simultaneously performed above a specified threshold level on the y-axis) that were plotted against each threshold level to determine the strength of the BD-MF relationship in control versus ant suppression plots across the multiple thresholds (i.e. every 1% increment in threshold value between 5% and 95% of function).”* L.638.

Some comments were left on the extended data files

Response 1.10: We thank the reviewer for pointing this out. These comments have now been removed.

Iridomyrmex purpureus appears to be missing from Table S3. Also, from Table 3, justification of these being dominant species is unclear: the abundance of the “dominant” species *Iridomyrmex purpureus* (missing, but plausibly dominant), *Pheidole amplapertensis* (data missing), *Tetramorium impressum* (3 in control and 11 in suppression plots), *Monomorium sordidum* (218, 240 – plausibly dominant), and, *Melophorus turneri* (425, 749, plausibly dominant).

Response 1.11: Our apologies, there seems to have been some confusion here. Characterization of species as ‘dominant’ at our site was based on site-wide sampling of ant communities before experimental suppression, and it is Table S2 (not Table S3) that shows the pre-treatment ant dominance data.

Table S3 shows non-target species responses to dominant ant removal, after the ant suppression treatments were imposed, which is why they the dominant species are missing.

We have now made this clearer in the manuscript by citing Table S2 and Table S3 (previously they were only cited in the captions to Extended Data Figure 1 and Extended Data Figure 4).

We have also modified the captions to Tables S2 and S3 to make it clearer what data are being presented. We now write:

Table S2. Incidence of species in pitfall trap samples across each experimental block in 2012 prior to ant suppression treatments being imposed. Data support the analysis of species dominance and trait group allocation for the species represented in Extended Data Figure 1.

*Table S3. Total abundances of non-target ant species sampled in pitfall traps in the year following suppression of the three dominant target species (Nov 2014 – Nov 2015). Data support the analysis of non-target species responses to dominant species removal shown in Extended Data Figure 4, so the target species *Iridomyrmex purpureus*, *Pheidole amplapertensis*, and *Tetramorium impressum* are not included here.*

Reviewer #2 (Remarks to the Author):

In this study, the authors use experimental removal of dominant ant species to test if the remaining species compensate the loss of ecological functions through functional redundancy effects. This question is both timely and touches on fundamental theory within the biodiversity-ecosystem functioning framework and therefore highly relevant. The authors manage to answer this theoretical question for their study system using an approach which is both elegantly designed and thoroughly planned and conducted. In addition, the authors did a very good job in documenting their approach, making their reasoning and decision process clear and in describing everything in a clear way. It was really a pleasure to read such a high quality manuscript.

Response 2.1: We thank the reviewer for the positive feedback about the importance of the work, and we appreciate the kind words about the quality of the manuscript.

I do have two suggestions for further improvement:

1. The abstract is a bit too abstract and technical in the results part. While it has to be concise and this is difficult with a theoretical background and counter-intuitive results, I think readers would benefit from simpler sentences and less detail. For example, the information on how much the trait dispersion was reduced is not necessary here. The sentence in line 21 is also very long and complicated.

Response 2.2: We thank the reviewer for these comments. As recommended, we have removed the technical detail about trait dispersion and split the sentence at line 21 to simplify interpretation.

2. The fact that functional performance was measured by actually looking at ant activity around baits (and was not only inferred through the abundance of certain traits within pitfall traps for example) is only mentioned "in passing" during the first part of the manuscript. Only when the reader arrives at the figure on granivory etc. does it become

clear that functions were observed in the field. I would recommend to mention this part of the study explicitly early in the manuscript as this is a really strong addition to the study overall.

Response 2.3: This is a good point. We now emphasize at line 86 that the functions were assessed using field observations.

Minor comments

Line 108: There is a full stop missing at the end of the sentence

Response 2.4: Thank you, we have corrected this error.

Line 120: Confidence areas are shown by shaded areas not by dashed lines

Response 2.5: Thank you, we have corrected this error.